# Infinite-Horizon Gaussian Processes

**Arno Solin**[*]
Aalto University
arno.solin@aalto.fi

**James Hensman**
PROWLER.io
james@prowler.io

**Richard E. Turner**
University of Cambridge
ret26@cam.ac.uk

## Abstract

Gaussian processes provide a flexible framework for forecasting, removing noise, and interpreting long temporal datasets. State space modelling (Kalman filtering) enables these non-parametric models to be deployed on long datasets by reducing the complexity to linear in the number of data points. The complexity is still cubic in the state dimension $m$ which is an impediment to practical application. In certain special cases (Gaussian likelihood, regular spacing) the GP posterior will reach a steady posterior state when the data are very long. We leverage this and formulate an inference scheme for GPs with general likelihoods, where inference is based on single-sweep EP (assumed density filtering). The infinite-horizon model tackles the cubic cost in the state dimensionality and reduces the cost in the state dimension $m$ to $\mathcal{O}(m^2)$ per data point. The model is extended to online-learning of hyperparameters. We show examples for large finite-length modelling problems, and present how the method runs in real-time on a smartphone on a continuous data stream updated at 100 Hz.

## 1  Introduction

Gaussian process (GP, [25]) models provide a plug & play interpretable approach to probabilistic modelling, and would perhaps be more widely applied if not for their associated computational complexity: naïve implementations of GPs require the construction and decomposition of a kernel matrix at cost $\mathcal{O}(n^3)$, where $n$ is the number of data. In this work, we consider GP time series (*i.e.* GPs with one input dimension). In this case, construction of the kernel matrix can be avoided by exploiting the (approximate) Markov structure of the process and re-writing the model as a linear Gaussian state space model, which can then be solved using Kalman filtering (see, *e.g.*, [27]). The Kalman filter costs $\mathcal{O}(m^3 n)$, where $m$ is the dimension of the state space. We propose the Infinite-Horizon GP approximation (IHGP), which reduces the cost to $\mathcal{O}(m^2 n)$.

As $m$ grows with the number of kernel components in the GP prior, this cost saving can be significant for many GP models where $m$ can reach hundreds. For example, the automatic statistician [6] searches for kernels (on 1D datasets) using sums and products of kernels. The summing of two kernels results in the concatenation of the state space (sum of the $m$s) and a product of kernels results in the Kronecker sum of their statespaces (product of $m$s). This quickly results in very high state dimensions; we show results with a similarly constructed kernel in our experiments.

We are concerned with real-time processing of long (or streaming) time-series with short and long length-scale components, and non-Gaussian noise/likelihood and potential non-stationary structure. We show how the IHGP can be applied in the streaming setting, including efficient estimation of the marginal likelihood and associated gradients, enabling on-line learning of hyper (kernel) parameters. We demonstrate this by applying our approach to a streaming dataset of two million points, as well as providing an implementation of the method on an iPhone, allowing on-line learning of a GP model of the phone's acceleration.

---

[*]This work was undertaken whilst AS was a Visiting Research Fellow with University of Cambridge.

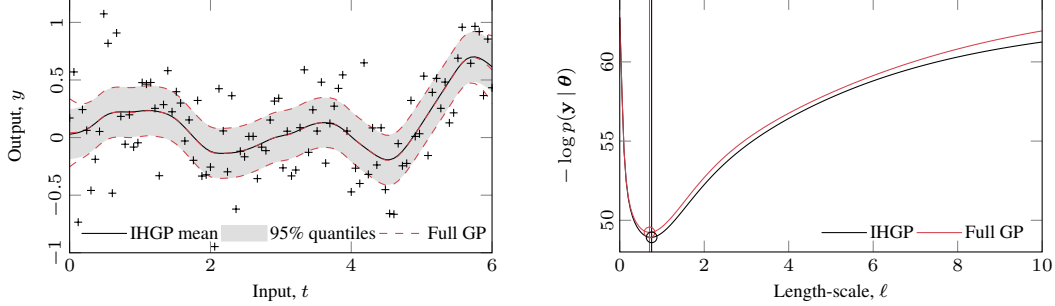

Figure 1: (Left) GP regression with $n = 100$ observations and a Matérn covariance function. The IHGP is close to exact far from boundaries, where the constant marginal variance assumption shows. (Right) Hyperparameters $\boldsymbol{\theta} = (\sigma_{\mathrm{n}}^2, \sigma^2, \ell)$ optimised independently for both models.

For data where a Gaussian noise assumption may not be appropriate, many approaches have been proposed for approximation (see, *e.g.*, [21] for an overview). Here we show how to combine Assumed Density Filtering (ADF, a.k.a. single-sweep Expectation Propagation, EP [5, 12, 19]) with the IHGP. We are motivated by the application to Log-Gaussian Cox Processes (LGCP, [20]). Usually the LGCP model uses binning to avoid a doubly-intractable model; in this case it is desirable to have more bins in order to capture short-lengthscale effects, leading to more time points. Additionally, the desire to capture long-and-short-term effects means that the state space dimension $m$ can be large. We show that our approach is effective on standard benchmarks (coal-mining disasters) as well as a much larger dataset (airline accidents).

The structure of the paper is as follows. Sec. 2 covers the necessary background and notation related to GPs and state space solutions. Sec. 3 leverages the idea of steady-state filtering to derive IHGP. Sec. 4 illustrates the approach on several problems, and the supplementary material contains additional examples and a nomenclature for easier reading. Code implementations in MATLAB/C++/Objective-C and video examples of real-time operation are available at https://github.com/AaltoML/IHGP.

## 2 Background

We are concerned with GP models [25] admitting the form: $f(t) \sim \mathrm{GP}(\mu(t), \kappa(t, t'))$ and $\mathbf{y} \,|\, \mathbf{f} \sim \prod_{i=1}^{n} p(y_i \,|\, f(t_i))$, where the data $\mathcal{D} = \{(t_i, y_i)\}_{i=1}^{n}$ are input–output pairs, $\mu(t)$ the mean function, and $\kappa(t, t')$ the covariance function of the GP prior. The likelihood factorizes over the observations. This family covers many standard modelling problems, including regression and classification tasks. Without loss of generality, we present the methodology for zero-mean ($\mu(t) := 0$) GP priors. We approximate posteriors of the form (see [24] for an overview):

$$q(\mathbf{f} \,|\, \mathcal{D}) = \mathrm{N}(\mathbf{f} \,|\, \mathbf{K}\boldsymbol{\alpha}, (\mathbf{K}^{-1} + \mathbf{W})^{-1}), \tag{1}$$

where $K_{i,j} = \kappa(t_i, t_j)$ is the prior covariance matrix, $\boldsymbol{\alpha} \in \mathbb{R}^n$, and the (likelihood precision) matrix is diagonal, $\mathbf{W} = \mathrm{diag}(\mathbf{w})$. Elements of $\mathbf{w} \in \mathbb{R}^n$ are non negative for log-concave likelihoods. The predictive mean and marginal variance for a test input $t_*$ is $\mu_{\mathrm{f},*} = \mathbf{k}_*^{\mathsf{T}} \boldsymbol{\alpha}$ and $\sigma_{\mathrm{f},*}^2 = k_{**} - \mathbf{k}_*^{\mathsf{T}} (\mathbf{K} + \mathbf{W}^{-1})^{-1} \mathbf{k}_*$. A probabilistic way of learning the hyperparameters $\boldsymbol{\theta}$ of the covariance function (such as magnitude and scale) and the likelihood model (such as noise scale) is by maximizing the (log) marginal likelihood function $p(\mathbf{y} \,|\, \boldsymbol{\theta})$ [25].

Numerous methods have been proposed for dealing with the prohibitive computational complexity of the matrix inverse in dealing with the latent function in Eq. (1). While general-purpose methods such as inducing input [4, 23, 30, 33], basis function projection [11, 17, 32], interpolation approaches [37], or stochastic approximations [10, 14] do not pose restrictions to the input dimensionality, they scale poorly in long time-series models by still needing to fill the extending domain (see discussion in [3]). For certain problems tree-structured approximations [3] or band-structured matrices can be leveraged. However, [8, 22, 26, 29] have shown that for one-dimensional GPs with high-order Markovian structure, an *optimal* representation (without approximations) is rewriting the GP in terms of a state space model and solving inference in *linear* time by sequential Kalman filtering methods. We will therefore focus on building upon the state space methodology.

## 2.1 State space GPs

In one-dimensional GPs (time-series) the data points feature the special property of having a natural ordering. If the GP prior itself admits a Markovian structure, the GP model can be reformulated as a state space model. Recent work has focused on showing how many widely used covariance function can be either exactly (*e.g.*, the half-integer Matérn class, polynomial, noise, constant) or approximately (*e.g.*, the squared-exponential/RBF, rational quadratic, periodic, *etc.*) converted into state space models. In continuous time, a simple dynamical system able to represent these covariance functions is given by the following linear time-invariant stochastic differential equation (see [28]):

$$\dot{\mathbf{f}}(t) = \mathbf{F}\,\mathbf{f}(t) + \mathbf{L}\,\mathbf{w}(t), \quad y_i \sim p(y_i \,|\, \mathbf{h}^\mathsf{T}\,\mathbf{f}(t_i)), \tag{2}$$

where $\mathbf{w}(t)$ is an $s$-dimensional white noise process, and $\mathbf{F} \in \mathbb{R}^{m \times m}$, $\mathbf{L} \in \mathbb{R}^{m \times s}$, $\mathbf{h} \in \mathbb{R}^{m \times 1}$ are the feedback, noise effect, and measurement matrices, respectively. The driving process $\mathbf{w}(t) \in \mathbb{R}^s$ is a multivariate white noise process with spectral density matrix $\mathbf{Q}_c \in \mathbb{R}^{s \times s}$. The initial state is distributed according to $\mathbf{f}_0 \sim \mathrm{N}(\mathbf{0}, \mathbf{P}_0)$. For discrete input values $t_i$, this translates into

$$\mathbf{f}_i \sim \mathrm{N}(\mathbf{A}_{i-1}\mathbf{f}_{i-1}, \mathbf{Q}_{i-1}), \quad y_i \sim p(y_i \,|\, \mathbf{h}^\mathsf{T}\mathbf{f}_i), \tag{3}$$

with $\mathbf{f}_0 \sim \mathrm{N}(\mathbf{0}, \mathbf{P}_0)$. The discrete-time dynamical model is solved through a matrix exponential $\mathbf{A}_i = \exp(\mathbf{F}\,\Delta t_i)$, where $\Delta t_i = t_{i+1} - t_i \geq 0$. For stationary covariance functions, $\kappa(t, t') = \kappa(t - t')$, the process noise covariance is given by $\mathbf{Q}_i = \mathbf{P}_\infty - \mathbf{A}_i\,\mathbf{P}_\infty\,\mathbf{A}_i^\mathsf{T}$. The stationary state (corresponding to the initial state $\mathbf{P}_0$) is distributed by $\mathbf{f}_\infty \sim \mathrm{N}(\mathbf{0}, \mathbf{P}_\infty)$ and the stationary covariance can be found by solving the Lyapunov equation $\dot{\mathbf{P}}_\infty = \mathbf{F}\,\mathbf{P}_\infty + \mathbf{P}_\infty\,\mathbf{F}^\mathsf{T} + \mathbf{L}\,\mathbf{Q}_c\,\mathbf{L}^\mathsf{T} = \mathbf{0}$. Appendix B shows an example of representing the Matérn ($\nu = 3/2$) covariance function as a state space model. Other covariance functions have been listed in [31].

## 2.2 Bayesian filtering

The closed-form solution to the linear Bayesian filtering problem—Eq. (3) with a Gaussian likelihood $\mathrm{N}(y_i \,|\, \mathbf{h}^\mathsf{T}\mathbf{f}_i, \sigma_\mathrm{n}^2)$—is known as the Kalman filter [27]. The interest is in the following marginal distributions: $p(\mathbf{f}_i \,|\, y_{1:i-1}) = \mathrm{N}(\mathbf{f}_i \,|\, \mathbf{m}_i^\mathrm{p}, \mathbf{P}_i^\mathrm{p})$ (predictive distribution), $p(\mathbf{f}_i \,|\, y_{1:i}) = \mathrm{N}(\mathbf{f}_i \,|\, \mathbf{m}_i^\mathrm{f}, \mathbf{P}_i^\mathrm{f})$ (filtering distribution), and $p(y_i \,|\, y_{1:i-1}) = \mathrm{N}(y_i \,|\, v_i, s_i)$ (decomposed marginal likelihood). The predictive state mean and covariance are given by $\mathbf{m}_i^\mathrm{p} = \mathbf{A}_i\,\mathbf{m}_{i-1}^\mathrm{f}$ and $\mathbf{P}_i^\mathrm{p} = \mathbf{A}_i\,\mathbf{P}_{i-1}^\mathrm{f}\,\mathbf{A}_i^\mathsf{T} + \mathbf{Q}_i$. The so called 'innovation' mean and variances $v_i$ and $s_i$ are

$$v_i = y_i - \mathbf{h}^\mathsf{T}\mathbf{m}_i^\mathrm{p} \qquad \text{and} \qquad s_i = \mathbf{h}^\mathsf{T}\mathbf{P}_i^\mathrm{p}\,\mathbf{h} + \sigma_\mathrm{n}^2. \tag{4}$$

The log marginal likelihood can be evaluated during the filter update steps by $\log p(\mathbf{y}) = -\sum_{i=1}^n \frac{1}{2}(\log 2\pi s_i + v_i^2/s_i)$. The filter mean and covariances are given by

$$\mathbf{k}_i = \mathbf{P}_i^\mathrm{p}\,\mathbf{h}/s_i, \qquad \mathbf{m}_i^\mathrm{f} = \mathbf{m}_{i-1}^\mathrm{p} + \mathbf{k}_i\,v_i, \qquad \mathbf{P}_i^\mathrm{f} = \mathbf{P}_i^\mathrm{p} - \mathbf{k}_i\,\mathbf{h}^\mathsf{T}\mathbf{P}_i^\mathrm{p}, \tag{5}$$

where $\mathbf{k}_i \in \mathbb{R}^m$ represents the filter *gain* term. In batch inference, we are actually interested in the so called *smoothing* solution, $p(\mathbf{f} \,|\, \mathcal{D})$ corresponding to marginals $p(\mathbf{f}_i \,|\, y_{1:n}) = \mathrm{N}(\mathbf{f}_i \,|\, \mathbf{m}_i^\mathrm{s}, \mathbf{P}_i^\mathrm{s})$. The smoother mean and covariance is solved by the backward recursion, from $i = n - 1$ backwards to 1:

$$\mathbf{m}_i^\mathrm{s} = \mathbf{m}_i^\mathrm{f} + \mathbf{G}_i\,(\mathbf{m}_{i+1}^\mathrm{s} - \mathbf{m}_{i+1}^\mathrm{p}), \qquad \mathbf{P}_i^\mathrm{s} = \mathbf{P}_i^\mathrm{f} + \mathbf{G}_i\,(\mathbf{P}_{i+1}^\mathrm{s} - \mathbf{P}_{i+1}^\mathrm{p})\,\mathbf{G}_i^\mathsf{T}, \tag{6}$$

where $\mathbf{G}_i = \mathbf{P}_i^\mathrm{f}\,\mathbf{A}_{i+1}^\mathsf{T}\,[\mathbf{P}_{i+1}^\mathrm{p}]^{-1}$ is the smoother *gain* at $t_i$. The computational complexity is clearly *linear* in the number of data $n$ (recursion repetitions), and *cubic* in the state dimension $m$ due to matrix–matrix multiplications, and the matrix inverse in calculation of $\mathbf{G}_i$.

## 3 Infinite-horizon Gaussian processes

We now tackle the cubic computational complexity in the state dimensionality by seeking infinite-horizon approximations to the Gaussian process. In Sec. 3.1 we revisit traditional steady-state Kalman filtering (for Gaussian likelihood, equidistant data) from quadratic filter design (see, *e.g.*, [18] and [7] for an introduction), and extend it to provide approximations to the marginal likelihood and its gradients. Finally, we present an infinite-horizon framework for non-Gaussian likelihoods.

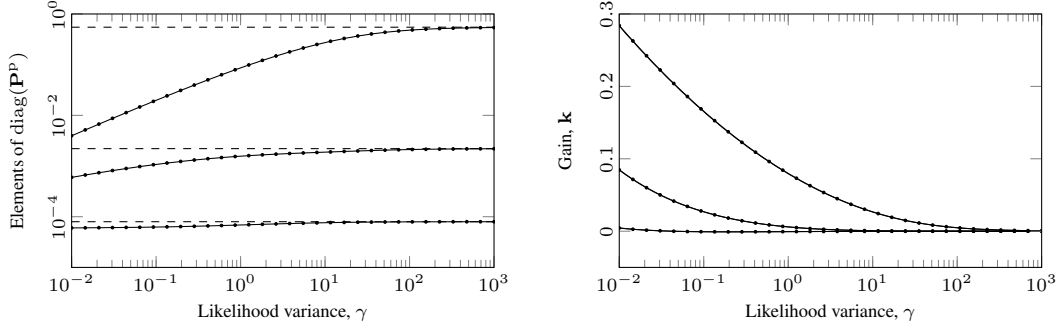

Figure 2: (Left) Interpolation of $\mathbf{P}^\mathrm{p}$ (dots solved, solid interpolated). The dashed lines show elements in $\mathbf{P}_\infty$ (prior stationary state covariance). (Right) The Kalman gain $\mathbf{k}$ evaluated for the $\mathbf{P}^\mathrm{p}$s.

## 3.1 Steady-state Kalman filter for $t \to \infty$

In steady-state Kalman filtering (see [7], Ch. 8.4, or [1], Ch. 4, for the traditional perspective) we assume $t \gg \ell_\mathrm{eff}$, where $\ell_\mathrm{eff}$ is the longest time scale in the covariance function, and equidistant observations in time ($\mathbf{A}_i := \mathbf{A}$ and $\mathbf{Q}_i := \mathbf{Q}$). After several $\ell_\mathrm{eff}$ (as $t \to \infty$), the filter gain converges to the stationary limiting Kalman filter gain $\mathbf{k}$. The resulting filter becomes time-invariant, which introduces approximation errors near the boundaries (*cf.* Fig. 1).

In practice, we seek a stationary filter state covariance (corresponding to the stationary Kalman gain) $\hat{\mathbf{P}}^\mathrm{f}$. Solving for this matrix thus corresponds to seeking a covariance that is equal between two consecutive filter recursions. Directly from the Kalman filtering forward prediction and update steps (in Eq. 5), we recover the recursion (by dropping dependency on the time step):

$$\hat{\mathbf{P}}^\mathrm{p} = \mathbf{A}\,\hat{\mathbf{P}}^\mathrm{p}\,\mathbf{A}^\mathsf{T} - \mathbf{A}\,\hat{\mathbf{P}}^\mathrm{p}\,\mathbf{h}\,(\mathbf{h}^\mathsf{T}\hat{\mathbf{P}}^\mathrm{p}\,\mathbf{h} + \sigma_\mathrm{n}^2)^{-1}\,\mathbf{h}^\mathsf{T}\hat{\mathbf{P}}^\mathrm{p}\,\mathbf{A}^\mathsf{T} + \mathbf{Q}. \tag{7}$$

This equation is of the form of a discrete algebraic Riccati equation (DARE, see, *e.g.*, [15]), which is a type of nonlinear matrix equation that often arises in the context of infinite-horizon optimal control problems. Since $\sigma_\mathrm{n}^2 > 0$, $\mathbf{Q}$ is P.S.D., and the associated state space model being both stabilizable and observable, the DARE has a unique stabilising solution for $\hat{\mathbf{P}}^\mathrm{p}$ that can be found either by iterating the Riccati equation or by matrix decompositions. The Schur method by Laub [16] solves the DARE in $\mathcal{O}(m^3)$, is numerically stable, and widely available in matrix libraries (Python `scipy.linalg.solve_discrete_are`, MATLAB Control System Toolbox `DARE`, see also SLICOT routine `SB02OD`).

The corresponding stationary gain is $\mathbf{k} = \hat{\mathbf{P}}^\mathrm{p}\,\mathbf{h}/(\mathbf{h}^\mathsf{T}\hat{\mathbf{P}}^\mathrm{p}\,\mathbf{h} + \sigma_\mathrm{n}^2)$. Re-deriving the filter recursion with the stationary gain gives a simplified iteration for the filter mean (the covariance is now time-invariant):

$$\hat{\mathbf{m}}_i^\mathrm{f} = (\mathbf{A} - \mathbf{k}\,\mathbf{h}^\mathsf{T}\mathbf{A})\,\hat{\mathbf{m}}_{i-1}^\mathrm{f} + \mathbf{k}\,y_i \quad \text{and} \quad \hat{\mathbf{P}}^\mathrm{f} = \hat{\mathbf{P}}^\mathrm{p} - \mathbf{k}\,\mathbf{h}^\mathsf{T}\hat{\mathbf{P}}^\mathrm{p}, \tag{8}$$

for all $i = 1, 2, \ldots, n$. This recursive iteration has a computational cost associated with one $m \times m$ matrix–vector multiplication, so the overall computational cost for the forward iteration is $\mathcal{O}(n\,m^2)$ (as opposed to the $\mathcal{O}(n\,m^3)$ in the Kalman filter).

**Marginal likelihood evaluation:** The approximative log marginal likelihood comes out as a by-product of the filter forward recursion: $\log p(\mathbf{y}) \approx -\frac{n}{2}\,\log 2\pi\hat{s} - \sum_{i=1}^n \hat{v}_i^2/(2\,\hat{s})$, where the stationary innovation covariance is given by $\hat{s} = \mathbf{h}^\mathsf{T}\hat{\mathbf{P}}^\mathrm{p}\,\mathbf{h} + \sigma_\mathrm{n}^2$ and the innovation mean by $\hat{v}_i = y_i - \mathbf{h}^\mathsf{T}\mathbf{A}\,\hat{\mathbf{m}}_{i-1}^\mathrm{f}$.

**Steady-state backward pass:** To obtain the complete infinite-horizon solution, we formally derive the solution corresponding to the smoothing distribution $p(\mathbf{f}_i \,|\, y_{1:n}) \approx \mathrm{N}(\mathbf{f}_i \,|\, \hat{\mathbf{m}}_i^\mathrm{s}, \hat{\mathbf{P}}^\mathrm{s})$, where $\hat{\mathbf{P}}$ is the stationary state covariance. Establishing the backward recursion does not require taking any additional limits, as the smoother gain is only a function of consecutive filtering steps. Re-deriving the backward pass in Equation (6) gives the time-invariant smoother gain and posterior state covariance

$$\mathbf{G} = \hat{\mathbf{P}}^\mathrm{f}\,\mathbf{A}^\mathsf{T}\,[\mathbf{A}\,\hat{\mathbf{P}}^\mathrm{f}\,\mathbf{A}^\mathsf{T} + \mathbf{Q}]^{-1} \quad \text{and} \quad \hat{\mathbf{P}}^\mathrm{s} = \mathbf{G}\,\hat{\mathbf{P}}^\mathrm{s}\,\mathbf{G}^\mathsf{T} + \hat{\mathbf{P}}^\mathrm{f} - \mathbf{G}\,(\mathbf{A}\,\hat{\mathbf{P}}^\mathrm{f}\,\mathbf{A}^\mathsf{T} + \mathbf{Q})\,\mathbf{G}^\mathsf{T}, \tag{9}$$

where $\hat{\mathbf{P}}^\mathrm{s}$ is implicitly defined in terms of the solution to a DARE. The backward iteration for the state mean: $\hat{\mathbf{m}}_i^\mathrm{s} = \hat{\mathbf{m}}_i^\mathrm{f} + \mathbf{G}\,(\hat{\mathbf{m}}_{i+1}^\mathrm{s} - \mathbf{A}\,\hat{\mathbf{m}}_i^\mathrm{f})$. Even this recursion scales as $\mathcal{O}(n\,m^2)$.

**Algorithm 1** Infinite-horizon Gaussian process (IHGP) inference. The GP prior is specified in terms of a state space model. After the setup cost on line 2, all operations are at most $\mathcal{O}(m^2)$.

1: **Input:** $\{y_i\}, \{\mathbf{A}, \mathbf{Q}, \mathbf{h}, \mathbf{P}_0\}, p(y \,|\, f)$              *targets, model, likelihood*
2: Set up $\mathbf{P}^{\mathrm{p}}(\gamma)$, $\mathbf{P}^{\mathrm{s}}(\gamma)$, and $\mathbf{G}(\gamma)$ for $\gamma_{1:K}$   *solve DAREs for a set of likelihood variances, cost $\mathcal{O}(K\,m^3)$*
3: $\mathbf{m}_0^{\mathrm{f}} \leftarrow \mathbf{0}$;    $\mathbf{P}_0^{\mathrm{p}} \leftarrow \mathbf{P}_0$;    $\gamma_0 = \infty$                          *initialize*
4: **for** $i = 1$ **to** $n$ **do**
5:      Evaluate $\mathbf{P}_i^{\mathrm{p}} \leftarrow \mathbf{P}^{\mathrm{p}}(\gamma_{i-1})$                     *find predictive covariance*
6:      $\tilde{\mu}_{\mathrm{f},i} \leftarrow \mathbf{h}^{\mathsf{T}} \mathbf{A} \, \mathbf{m}_{i-1}^{\mathrm{f}}$;    $\tilde{\sigma}_{\mathrm{f},i}^2 = \mathbf{h}^{\mathsf{T}} \mathbf{P}_i^{\mathrm{p}} \, \mathbf{h}$                     *latent*
7:      **if** Gaussian likelihood **then**
8:         $\eta_i \leftarrow y_i$;    $\gamma_i \leftarrow \sigma_{\mathrm{n},i}^2$           *if $\sigma_{\mathrm{n},i}^2 := \sigma_{\mathrm{n}}^2$, $\mathbf{k}_i$ and $\mathbf{P}_i^{\mathrm{f}}$ become time-invariant*
9:      **else**
10:         Match $\exp(\nu_i \, f_i - \tau_i \, f_i^2 / 2)\, \mathrm{N}(f_i \,|\, \tilde{\mu}_{\mathrm{f},i}, \tilde{\sigma}_{\mathrm{f},i}^2) \overset{\mathrm{mom}}{=} p(y_i \,|\, f_i)\, \mathrm{N}(f_i \,|\, \tilde{\mu}_{\mathrm{f},i}, \tilde{\sigma}_{\mathrm{f},i}^2)$    *match moments*
11:         $\eta_i \leftarrow \nu_i / \tau_i$;    $\gamma_i \leftarrow \tau_i^{-1}$                      *equivalent update*
12:      **end if**
13:      $\mathbf{k}_i \leftarrow \mathbf{P}_i^{\mathrm{p}} \, \mathbf{h} / (\tilde{\sigma}_{\mathrm{f},i}^2 + \gamma_i)$                            *gain*
14:      $\mathbf{m}_i^{\mathrm{f}} \leftarrow (\mathbf{A} - \mathbf{k}_i \, \mathbf{h}^{\mathsf{T}} \mathbf{A}) \, \mathbf{m}_{i-1}^{\mathrm{f}} + \mathbf{k}_i \, \eta_i$;    $\mathbf{P}_i^{\mathrm{f}} \leftarrow \mathbf{P}_i^{\mathrm{p}} - \mathbf{k}_i \, \gamma_i \, \mathbf{k}_i^{\mathsf{T}}$    *mean and covariance*
15: **end for**
16: $\mathbf{m}_n^{\mathrm{s}} \leftarrow \mathbf{m}_n^{\mathrm{f}}$;    $\mathbf{P}_n^{\mathrm{s}} \leftarrow \mathbf{P}^{\mathrm{s}}(\gamma_n)$                  *initialize backward pass*
17: **for** $i = n - 1$ **to** $1$ **do**
18:      $\mathbf{m}_i^{\mathrm{s}} \leftarrow \mathbf{m}_i^{\mathrm{f}} + \mathbf{G}(\gamma_i) \, (\mathbf{m}_{i+1}^{\mathrm{s}} - \mathbf{A} \, \mathbf{m}_i^{\mathrm{f}})$;    $\mathbf{P}_i^{\mathrm{s}} \leftarrow \mathbf{P}^{\mathrm{s}}(\gamma_i)$    *mean and covariance*
19: **end for**
20: **Return:** $\mu_{\mathrm{f},i} = \mathbf{h}^{\mathsf{T}} \mathbf{m}_i^{\mathrm{s}}$, $\sigma_{\mathrm{f},i}^2 = \mathbf{h}^{\mathsf{T}} \mathbf{P}_i^{\mathrm{s}} \, \mathbf{h}$, $\log p(\mathbf{y})$        *mean, variance, evidence*

## 3.2 Infinite-horizon GPs for general likelihoods

In IHGP, instead of using the true predictive covariance for propagation, we use the one obtained from the stationary state of a system with measurement noise fixed to the current measurement noise and regular spacing. The Kalman filter iterations can be used in solving approximate posteriors for models with general likelihoods in form of Eq. (1) by manipulating the innovation $v_i$ and $s_i$ (see [22]). We derive a generalization of the steady-state iteration allowing for time-dependent measurement noise and non-Gaussian likelihoods.

We re-formulate the DARE in Eq. (7) as an implicit function $\hat{\mathbf{P}}^{\mathrm{p}} : \mathbb{R}_+ \to \mathbb{R}^{m \times m}$ of the likelihood variance, 'measurement noise', $\gamma \in \mathbb{R}_+$:

$$\mathbf{P}^{\mathrm{p}}(\gamma) = \mathbf{A} \, \mathbf{P}^{\mathrm{p}}(\gamma) \, \mathbf{A}^{\mathsf{T}} - \mathbf{A} \, \mathbf{P}^{\mathrm{p}}(\gamma) \, \mathbf{h} \, (\mathbf{h}^{\mathsf{T}} \mathbf{P}^{\mathrm{p}}(\gamma) \, \mathbf{h} + \gamma)^{-1} \, \mathbf{h}^{\mathsf{T}} \mathbf{P}^{\mathrm{p}}(\gamma) \, \mathbf{A}^{\mathsf{T}} + \mathbf{Q}. \qquad (10)$$

The elements in $\mathbf{P}^{\mathrm{p}}$ are smooth functions in $\gamma$, and we set up an interpolation scheme—inspired by Wilson and Nickisch [37] who use cubic convolutional interpolation [13] in their KISS-GP framework—over a log-spaced one-dimensional grid of $K$ points in $\gamma$ for evaluation of $\hat{\mathbf{P}}^{\mathrm{p}}(\gamma)$. Fig. 2 shows results of $K = 32$ grid points (as dots) over $\gamma = 10^{-2}, \ldots, 10^3$ (this grid is used throughout the experiments). In the limit of $\gamma \to \infty$ the measurement has no effect, and the predictive covariance returns to the stationary covariance of the GP prior (dashed). Similarly, the corresponding gain terms $\mathbf{k}$ show the gains going to zero in the same limit. We set up a similar interpolation scheme for evaluating $\mathbf{G}(\gamma)$ and $\mathbf{P}^{\mathrm{s}}(\gamma)$ following Eq. (9). Now, solving the DAREs and the smoother gain has been replaced by computationally cheap (one-dimensional) kernel interpolation.

Alg. 1 presents the recursion in IHGP inference by considering a locally steady-state GP model derived from the previous section. As can be seen in Sec. 3.1, the predictive state on step $i$ only depends on $\gamma_{i-1}$. For non-Gaussian inference we set up an EP [5, 12, 19] scheme which only requires one forward pass (assumed density filtering, see also *unscented* filtering [27]), and is thus well suited for streaming applications. We match the first two moments of $p(y_i \,|\, f_i)$ and $\exp(\tau \, f_i - \nu \, f_i^2 / 2)$ w.r.t. latent values $\mathrm{N}(f_i \,|\, \tilde{\mu}_{\mathrm{f},i}, \tilde{\sigma}_{\mathrm{f},i}^2)$ (denoted by $\bullet \overset{\mathrm{mom}}{=} \bullet$, implemented by quadrature). The steps of the backward pass are also only dependent on the local steady-state model, thus evaluated in terms of $\gamma_i$s.

Missing observations correspond to $\gamma_i = \infty$, and the model could be generalized to non-equidistant time sampling by the scheme in Nickisch et al. [22] for calculating $\mathbf{A}(\Delta t_i)$ and $\mathbf{Q}(\Delta t_i)$.

Table 1: Mean absolute error of IHGP w.r.t. SS, negative log-likelihoods, and running times. Mean over 10 repetitions reported; $n = 1000$.

| | Regression | Count data | Classification | |
|---|---|---|---|---|
| Likelihood | Gaussian | Poisson | Logit | Probit |
| MAE $\mathbb{E}[f(t_*)]$ | 0.0095 | 0.0415 | 0.0741 | 0.0351 |
| MAE $\mathbb{V}[f(t_*)]$ | 0.0008 | 0.0024 | 0.0115 | 0.0079 |
| NLL-FULL | 1452.5 | 2645.5 | 618.9 | 614.4 |
| NLL-SS | 1452.5 | 2693.5 | 617.5 | 613.9 |
| NLL-IHGP | 1456.0 | 2699.3 | 625.1 | 618.2 |
| $t_{\text{full}}$ | 0.18 s | 6.17 s | 11.78 s | 9.93 s |
| $t_{\text{ss}}$ | 0.04 s | 0.13 s | 0.13 s | 0.11 s |
| $t_{\text{IHGP}}$ | 0.01 s | 0.14 s | 0.13 s | 0.10 s |

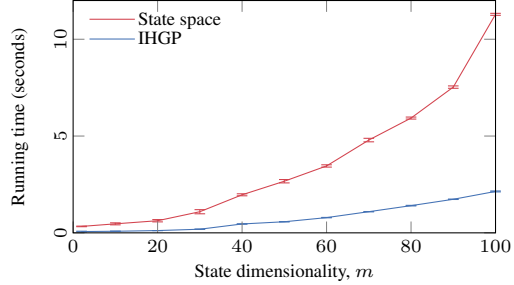

Figure 3: Empirical running time comparison for GP regression on $n = 10{,}000$ data points. Maximum RMSE in IHGP $\mathbb{E}[f(t_*)] < 0.001$.

## 3.3 Online hyperparameter estimation

Even though IHGP can be used in a batch setting, it is especially well suited for continuous data streams. In such applications, it is not practical to require several iterations over the data for optimising the hyperparameters—as new data would arrive before the optimisation terminates. We propose a practical extension of IHGP for online estimation of hyperparameters $\boldsymbol{\theta}$ by leveraging that *(i)* new batches of data are guaranteed to be available from the stream, *(ii)* IHGP only requires seeing each data point once for evaluating the marginal likelihood and its gradient, *(iii)* data can be non-stationary, requiring the hyperparameters to adapt.

We formulate the hyperparameter optimisation problem as an incremental gradient descent (*e.g.*, [2]) resembling stochastic gradient descent, but without the assumption of finding a stationary optimum. Starting from some initial set of hyperparameters $\boldsymbol{\theta}_0$, for each new (mini) batch $j$ of data $\mathbf{y}^{(j)}$ in a window of size $n_{\text{mb}}$, iterate

$$\boldsymbol{\theta}_j = \boldsymbol{\theta}_{j-1} + \eta \, \nabla \log p(\mathbf{y}^{(j)} \,|\, \boldsymbol{\theta}_{j-1}), \tag{11}$$

where $\eta$ is a learning-rate (step-size) parameter, and the gradient of the marginal likelihood is evaluated by the IHGP forward recursion. In a vanilla GP the windowing would introduce boundary effect due to growing marginal variance towards the boundaries, while in IHGP no edge effects are present as the data stream is seen to continue beyond any boundaries (*cf.* Fig. 1).

## 4 Experiments

We provide extensive evaluation of the IHGP both in terms of simulated benchmarks and four real-world experiments in batch and online modes.

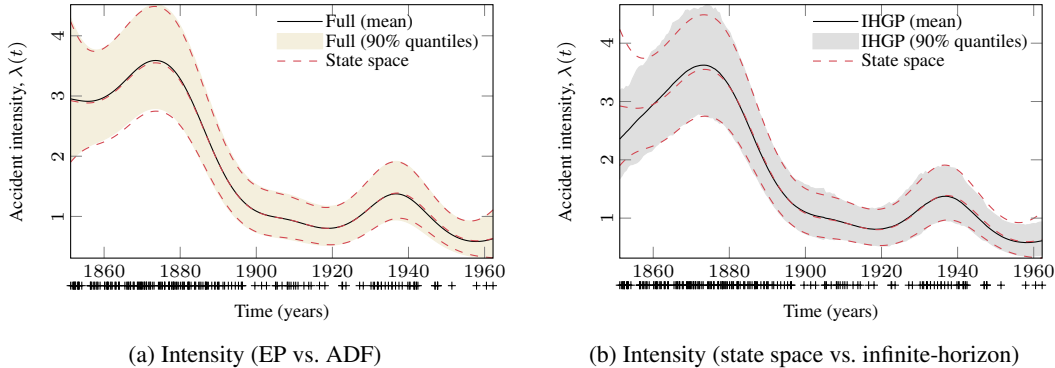

(a) Intensity (EP vs. ADF)

(b) Intensity (state space vs. infinite-horizon)

Figure 4: A small-scale comparison study on the coal mining accident data (191 accidents in $n = 200$ bins). The data set is sufficiently small that full EP with naïve handling of the latent function can be conducted. Full EP is shown to work similarly as ADF (single-sweep EP) by state space modelling. We then compare ADF on state space (exact handling of the latent function) to ADF with the IHGP.

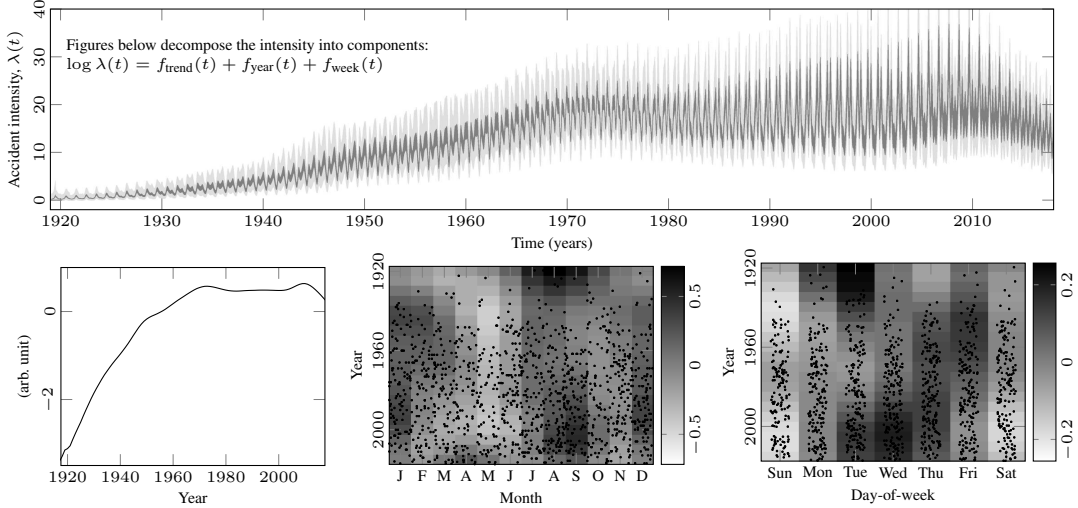

Figure 5: Explanatory analysis of the aircraft accident data set (1210 accidents predicted in $n = 35{,}959$ daily bins) between years 1919–2018 by a log-Gaussian Cox process (Poisson likelihood).

## 4.1 Experimental validation

In the toy examples, the data were simulated from $y_i = \mathrm{sinc}(x_i - 6) + \varepsilon_i$, $\varepsilon_i \sim \mathrm{N}(0, 0.1)$ (see Fig. 1 for a visualization). The same function with thresholding was used in the classification examples in the Appendix. Table 1 shows comparisons for different log-concave likelihoods over a simulated data set with $n = 1000$. Example functions can be seen in Fig. 1 and Appendix E. The results are shown for a Matérn ($\nu = 3/2$) with a full GP (naïve handling of latent, full EP as in [24]), state space (SS, exact state space model, ADF as in [22]), and IHGP. With $m$ only 2, IHGP is not faster than SS, but approximation errors remain small. Fig. 3 shows experimental results for the computational benefits in a regression study, with state dimensionality $m = 2, \ldots, 100$. Experiments run in Mathworks MATLAB (R2017b) on an Apple MacBook Pro (2.3 GHz Intel Core i5, 16 Gb RAM). Both methods have linear time complexity in the number of data points, so the number of data points is fixed to $n = 10{,}000$. The GP prior is set up as an increasing-length sum of Matérn ($\nu = 3/2$) kernels with different characteristic length-scales. The state space scheme follows $\mathcal{O}(m^3)$ and IHGP is $\mathcal{O}(m^2)$.

## 4.2 Log-Gaussian Cox processes

A log Gaussian Cox process is an inhomogeneous Poisson process model for count data. The unknown intensity function $\lambda(t)$ is modelled with a log-Gaussian process such that $f(t) = \log \lambda(t)$. The likelihood of the unknown function $f$ is $p(\{t_j\} \,|\, f) = \exp(-\int \exp(f(t)) \, \mathrm{d}t + \sum_{j=1}^{N} f(t_j))$. The likelihood requires non-trivial integration over the exponentiated GP, and thus instead the standard approach [20] is to consider locally constant intensity in subregions by discretising the interval into bins. This approximation corresponds to having a Poisson model for each bin. The likelihood becomes $p(\{t_j\} \,|\, f) \approx \prod_{i=1}^{n} \mathrm{Poisson}(y_i(\{t_j\}) \,|\, \exp(f(\hat{t}_i)))$, where $\hat{t}_i$ is the bin coordinate and $y_i$ the number of data points in it. This model reaches posterior consistency in the limit of bin width going to zero [34]. Thus it is expected that the accuracy improves with tighter binning.

**Coal mining disasters dataset:** The data (available, *e.g.*, in [35]) contain the dates of 191 coal mine explosions that killed ten or more people in Britain between years 1851–1962, which we discretize into $n = 200$ bins. We use a GP prior with a Matérn ($\nu = 5/2$) covariance function that has an exact state space representation (state dimensionality $m = 3$) and thus no approximations regarding handling the latent are required. We optimise the characteristic length-scale and magnitude hyperparameters w.r.t. marginal likelihood in each model. Fig. 4 shows that full EP and state space ADF produce almost equivalent results, and IHGP ADF and state space ADF produce similar results. In IHGP the edge effects are clear around 1850–1860.

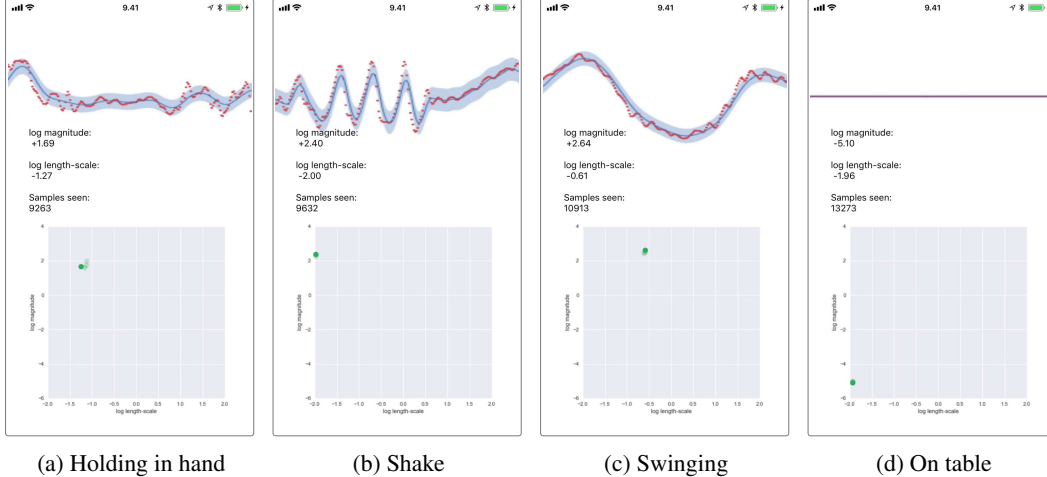

| (a) Holding in hand | (b) Shake | (c) Swinging | (d) On table |

Figure 6: Screenshots of online adaptive IHGP running in real-time on an iPhone. The lower plot shows current hyperparameters (measurement noise is fixed to $\sigma_{\mathrm{n}}^2 = 1$ for easier visualization) of the prior covariance function, with a trail of previous hyperparameters. The top part shows the last 2 seconds of accelerometer data (red), the GP mean, and 95% quantiles. The refresh rate for updating the hyperparameters and re-prediction is 10 Hz. Video examples are in the supplementary material.

**Airline accident dataset:** As a more challenging regression problem we explain the time-dependent intensity of accidents and incidents of commercial aircraft. The data [22] consists of dates of 1210 incidents over the time-span of years 1919–2017. We use a bin width of one day and start from year 1900 ensure no edge effects ($n = 43{,}099$), and a prior covariance function (similar to [6, 36])

$$\kappa(t, t') = \kappa_{\mathrm{Mat.}}^{\nu=5/2}(t, t') + \kappa_{\mathrm{per}}^{1\,\mathrm{year}}(t, t')\,\kappa_{\mathrm{Mat.}}^{\nu=3/2}(t, t') + \kappa_{\mathrm{per}}^{1\,\mathrm{week}}(t, t')\,\kappa_{\mathrm{Mat.}}^{\nu=3/2}(t, t') \tag{12}$$

capturing a trend, time-of-year variation (with decay), and day-of-week variation (with decay). This model has a state space representation of dimension $m = 3 + 28 + 28 = 59$. All hyperparameters (except time periods) were optimised w.r.t. marginal likelihood. Fig. 5 shows that we reproduce the time-of-year results from [22] and additionally recover a high-frequency time-of-week effect.

### 4.3 Electricity consumption

We do explorative analysis of electricity consumption for one household [9] recorded every minute (in log kW) over 1,442 days ($n = 2{,}075{,}259$, with 25,979 missing observations). We assign the model a GP prior with a covariance function accounting for slow variation and daily periodicity (with decay). We fit a GP to the entire data with 2M data points by optimising the hyperparameters w.r.t. marginal likelihood (results shown in Appendix F) using BFGS. Total running time 624 s.

The data is, however, inherently non-stationary due to the long time-horizon, where use of electricity has varied. We therefore also run IHGP online in a rolling-window of 10 days ($n_{\mathrm{mb}} = 14{,}400$, $\eta = 0.001$, window step size of 1 hr) and learn the hyperparameters online during the 34,348 incremental gradient steps (evaluation time per step $0.26\pm0.05$ s). This leads to a non-stationary adaptive GP model which, *e.g.*, learns to dampen the periodic component when the house is left vacant for days. Results shown in Appendix F in the supplement.

### 4.4 Real-time GPs for adaptive model fitting

In the final experiment we implement the IHGP in C++ with wrappers in Objective-C for running as an app on an Apple iPhone 6s (iOS 11.3). We use the phone accelerometer $x$ channel (sampled at 100 Hz) as an input and fit a GP to a window of 2 s with Gaussian likelihood and a Matérn ($\nu = {}^{3}/_{2}$) prior covariance function. We fix the measurement noise to $\sigma_{\mathrm{n}}^2 = 1$ and use separate learning rates $\boldsymbol{\eta} = (0.1, 0.01)$ in online estimation of the magnitude scale and length-scale hyperparameters. The GP is re-estimated every 0.1 s. Fig. 6 shows examples of various modes of data and how the GP has adapted to it. A video of the app in action is included in the web material together with the codes.

# 5 Discussion and conclusion

We have presented Infinite-Horizon GPs, a novel approximation scheme for state space Gaussian processes, which reduces the time-complexity to $\mathcal{O}(m^2n)$. There is a clear intuition to the approximation: As widely known, in GP regression the posterior marginal variance only depends on the distance between observations, and the likelihood variance. If both these are fixed, and $t$ is larger than the largest length-scale in the prior, the posterior marginal variance reaches a stationary state. The intuition behind IHGP is that for every time instance, we adapt to the current likelihood variance, discard the Markov-trail, and start over by adapting to the current steady-state marginal posterior distribution.

This approximation scheme is important especially in long (number of data in the thousands–millions) or streaming ($n$ growing without limit) data, and/or the GP prior has several components ($m$ large). We showed examples of regression, count data, and classification tasks, and showed how IHGP can be used in interpreting non-stationary data streams both off-line (Sec. 4.3) and on-line (Sec. 4.4).

**Acknowledgments**

We thank the anonymous reviewers as well as Mark Rowland and Will Tebbutt for their comments on the manuscript. AS acknowledges funding from the Academy of Finland (grant number 308640).

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
