[Supplementary Material]

# Supplementary Material for
# Infinite-horizon Gaussian processes

## A Nomenclature

In order of appearance. Vectors bold-face small letters, matrices bold-face capital letters.

| Symbol | Description |
|---|---|
| $n$ | Number of (training) data points |
| $m$ | State dimensionality |
| $t \in \mathbb{R}$ | Time (input) |
| $i$ | (Typically) Time index, $t_i$ |
| $y$ | Observation (output) |
| $\mathbf{y} \in \mathbb{R}^n$ | Collection of outputs, $(y_1, y_2, \ldots, y_n)$ |
| $\kappa(t, t')$ | Covariance function (kernel) |
| $\mu(t)$ | Mean function |
| $\boldsymbol{\theta}$ | Vector of model (hyper) parameters |
| $\sigma_{\mathrm{n}}^2$ | Measurement noise variance |
| $\ell$ | Characteristic length-scale |
| $\mathbf{K} \in \mathbb{R}^{n \times n}$ | Covariance (Gram) matrix, $K_{i,j} = \kappa(t_i, t_j)$ |
| $\mathbf{w} \in \mathbb{R}^n$ | Likelihood precision matrix diagonal |
| $f(t) : \mathbb{R} \to \mathbb{R}$ | Latent function |
| $\mathbf{f}$ | Vector of evaluated latent, $(f(t_1), f(t_2), \ldots, f(t_n))$ |
| $f_i$ | Element in $\mathbf{f}$ |
| $\mathbf{f}(t) : \mathbb{R} \to \mathbb{R}^m$ | Vector-valued latent function, $f(t) = \mathbf{h}^{\mathsf{T}} \mathbf{f}(t)$ |
| $\mathbf{f}_i \in \mathbb{R}^m$ | The state variable, $\mathbf{f}_i = \mathbf{f}(t_i)$ and $\mathbf{f}_i \sim \mathrm{N}(\mathbf{m}_i, \mathbf{P}_i)$ |
| $\mathbf{F} \in \mathbb{R}^{m \times m}$ | Feedback matrix (continuous-time model) |
| $\mathbf{L} \in \mathbb{R}^{m \times s}$ | Noise effect matrix (continuous-time model) |
| $\mathbf{Q}_{\mathrm{c}} \in \mathbb{R}^{s \times s}$ | Driving white noise spectral density (continuous-time model) |
| $\mathbf{h} \in \mathbb{R}^m$ | Measurement model |
| $\mathbf{A} \in \mathbb{R}^{m \times m}$ | Dynamic model (discrete-time model) |
| $\mathbf{Q} \in \mathbb{R}^{m \times m}$ | Process noise covariance (discrete-time model) |
| $\mathbf{P}_\infty \in \mathbb{R}^{m \times m}$ | Stationary state covariance (prior) |
| $\mathbf{m}_i \in \mathbb{R}^{m \times m}$ | State mean |
| $\mathbf{P}_i \in \mathbb{R}^{m \times m}$ | State covariance |
| $\mathbf{k} \in \mathbb{R}^m$ | Kalman gain |
| $\mathbf{G} \in \mathbb{R}^{m \times m}$ | Smoother gain |
| $v_i$ | Innovation mean |
| $s_i$ | Innovation variance |
| $\bullet^{\mathrm{p}}$ | Superscript 'p' denotes predictive quantities |
| $\bullet^{\mathrm{f}}$ | Superscript 'f' denotes filtering quantities |
| $\bullet^{\mathrm{s}}$ | Superscript 's' denotes smoothing quantities |
| $\hat{\bullet}$ | The hat denotes steady-state approximation quantities |
| $\gamma \in \mathbb{R}_+$ | Likelihood variance |
| $\eta$ | Learning rate |

## B   Example of a Matérn ($\nu = {}^3/{}_2$) covariance function

Consider the Matérn covariance function with smoothness $\nu = 3/2$, for which the processes are continuous and once differentiable:

$$\kappa_{\text{Mat.}}(t, t') = \sigma^2 \left( 1 + \frac{\sqrt{3}\,|t - t'|}{\ell} \right) \exp\left( -\frac{\sqrt{3}\,|t - t'|}{\ell} \right). \tag{13}$$

It has the SDE representation [8]

$$\mathbf{F} = \begin{pmatrix} 0 & 1 \\ -\lambda^2 & -2\lambda \end{pmatrix}, \quad \mathbf{L} = \begin{pmatrix} 0 \\ 1 \end{pmatrix}, \quad \mathbf{P}_\infty = \begin{pmatrix} \sigma^2 & 0 \\ 0 & \lambda^2\sigma^2 \end{pmatrix}, \quad \text{and} \quad \mathbf{h} = \begin{pmatrix} 1 \\ 0 \end{pmatrix}, \tag{14}$$

where $\lambda = \sqrt{3}/\ell$. The spectral density of the Gaussian white noise process $w(t)$ is $Q_{\text{c}} = 4\lambda^3\sigma^2$. For higher-order half-integer Matérn covariance functions, the state dimensionality follows the smoothness parameter, $m = \nu + 1/2$.

## C   Forward derivatives for efficient log likelihood gradient evaluation

The recursion for evaluating the derivatives of the log marginal likelihood can be derived by differentiating the steady-state recursions. As the equation for the stationary predictive covariance is given by the DARE:

$$\hat{\mathbf{P}}^{\text{p}} = \mathbf{A}\,\hat{\mathbf{P}}^{\text{p}}\,\mathbf{A}^{\mathsf{T}} - \mathbf{A}\,\hat{\mathbf{P}}^{\text{p}}\,\mathbf{h}\,(\mathbf{h}^{\mathsf{T}}\hat{\mathbf{P}}^{\text{p}}\,\mathbf{h} + \sigma_{\text{n}}^2)^{-1}\,\mathbf{h}^{\mathsf{T}}\hat{\mathbf{P}}^{\text{p}}\,\mathbf{A}^{\mathsf{T}} + \mathbf{Q}. \tag{15}$$

In order to evaluate the derivatives with respect to hyperparameters, the stationary covariance $\hat{\mathbf{P}}^{\text{p}}$ must be differentiated. In practice the model matrices $\mathbf{A}$ and $\mathbf{Q}$ are functions of the hyperparameter values $\boldsymbol{\theta}$ as is the measurement noise variance $\sigma_{\text{n}}^2$.

Differentiating gives:

$$\partial\hat{\mathbf{P}}^{\text{p}} = (\mathbf{A} - \mathbf{B}\,\mathbf{h}^{\mathsf{T}})\,\partial\hat{\mathbf{P}}^{\text{p}}\,(\mathbf{A} - \mathbf{B}\,\mathbf{h}^{\mathsf{T}})^{\mathsf{T}} + \mathbf{C}, \tag{16}$$

where $\mathbf{B} = \mathbf{A}\,\hat{\mathbf{P}}^{\text{p}}\,\mathbf{h}\,(\mathbf{h}^{\mathsf{T}}\hat{\mathbf{P}}^{\text{p}}\,\mathbf{h} + \sigma_{\text{n}}^2)^{-1}$ and $\mathbf{C} = \partial\mathbf{A}\,\hat{\mathbf{P}}^{\text{p}}\,\mathbf{A}^{\mathsf{T}} + \mathbf{A}\,\hat{\mathbf{P}}^{\text{p}}\,\partial\mathbf{A}^{\mathsf{T}} - \partial\mathbf{A}\,\hat{\mathbf{P}}^{\text{p}}\,\mathbf{h}\,\mathbf{B}^{\mathsf{T}} - \mathbf{B}\,\mathbf{h}^{\mathsf{T}}\hat{\mathbf{P}}^{\text{p}}\,\partial\mathbf{A}^{\mathsf{T}} + \mathbf{B}\,\partial\sigma_{\text{n}}^2\,\mathbf{B}^{\mathsf{T}} + \partial\mathbf{Q}$.

Equation (16) is also a DARE, which means that a DARE needs to be solved for each hyperparameter. However, after this initial cost evaluating the recursion for calculating the gradient of the negative log marginal likelihood is simply a matter of the following operations:

$$\nabla \log p(\mathbf{y}\,|\,\boldsymbol{\theta}) = -\frac{n}{2\,\hat{s}}\,\nabla\hat{s} - \sum_i \left[ \frac{\hat{v}_i}{\hat{s}}\,\nabla\hat{v}_i - \frac{\hat{v}_i^2}{2\,\hat{s}_i^2}\,\nabla\hat{s}_i \right], \tag{17}$$

where the recursion only has to propagate $\partial\mathbf{m}_i$ over steps for evaluating $\nabla\hat{s}_i$. The gradient can be evaluated very efficiently just as a matter of two additional $m^2$ matrix–vector multiplications per time step. This is different from the complete state space evaluations, where calculating the derivatives becomes costly as the entire Kalman filter needs to be differentiated.

## D   Stabilisation of the forward and backward gains

We have included a figure (Fig. 7) showing the quick stabilisation of the gains in running the toy experiment in Fig. 1. Even though the data is too small to be practical for IHGP, the edge-effects are not severe. For larger data sets, the likelihood curves in Fig. 1 keep approaching each others.

## E   Classification examples

We include two additional figures showing results for classification examples using simulated data. Fig. 8 shows the results.

(a) Forward gain

(b) Backward gain

Figure 7: Example of how the gain terms stabilize over the time span of 100 samples. The solid lines are the true gains and dashed lines the stabilizing infinite-horizon gains. These are the gains for the results in Fig. 1.

(a) Classification (logit)

(b) Classification (probit)

Figure 8: Two examples of IHGP classification on toy data (thresholded sinc function) with a Matérn ($\nu = {}^3/_2$) GP prior. The figure shows results (the mean and 95% quantiles squashed through the link function) for a full GP (naïve handling of latent, full EP inference), state space (exact state space inference of latent, ADF inference), and IHGP. The hyperparameters of the covariance function were optimised (w.r.t. marginal likelihood) independently using each model.

# F Electricity example

In the electricity consumption example we aim to explain the underlying process (occupancy and living rhythm) that generates the electricity consumption in the household.

We first perform GP batch regression with a GP prior with the covariance function

$$\kappa(t, t') = \kappa_{\text{Mat.}}^{\nu=3/2}(t, t') + \kappa_{\text{per}}^{1\,\text{day}}(t, t')\,\kappa_{\text{Mat.}}^{\nu=3/2}(t, t'), \tag{18}$$

where the first component captures the short or long-scale trend variation, and the second component is a periodic model that aims to capture the time of day variation (with decay, a long length-scale Matérn). In order not to over-fit, we fix the measurement noise variance and the length-scale of the multiplicative Matérn component. We optimised the remaining four hyperparameters with respect to marginal likelihood. The values are visualized in Fig. 9 with dashed lines. Total running time 624 s on the MacBook Pro used in all experiments.

As the stationary model is clearly an over-simplification of the modelling problem, we also apply IHGP in an online setting in finding the hyperparameters. Fig. 9 shows the adapted hyperparameter time-series over the entire time-range.

We have selected three 10-day windows (with 14,400 observations each) to highlight that the model manages to capture the changes in the data. Subfigure (a) shows the (noisy) daily variation with a clear periodic structure. In (b) the electricity consumption has been small for several days and the magnitude of both components has dropped. Furthermore, the periodic model has increased its length-scale to effectively turn itself off. In (c) the predictive capability of the model shows and captures the daily variation even though there has been a fault in the data collection.

(a) Typical daily rhythm

(b) House vacant

(c) Missing data

(d) Learned hyperparameters over the time-range

Figure 9: Results for explorative analysis of electricity consumption data over 1,442 days with one-minute resolution ($n > 2$M). (d) The batch optimized hyperparameters values shown by dashed lines, the results for IHGP with adaptation (solid) adapt to changing circumstances. (a)–(c) show three 10-day windows where the model has adapted to different modes of electricity consumption. Data shown by dots, predictive mean and 95% quantiles shown by the solid line and shaded regions.