[Reviews · NeurIPS 2018]

Reviewer 1



This submission proposes to use the stationary distribution of the stochastic function under the GP model for the prediction of streaming data. The required matrices follow the DARE, and the authors propose to solve for these using standard matrix libraries. The method is extended to non-Gaussian likelihoods. Hyper-parameter adjustments based on mini-batches are proposed. Experimental results on four real data sets are given. [Quality] This paper gives a new approximation approach to the state-space formulation of Gaussian processes, and it is especially useful for streaming data using processes with comparatively shorter length scales. However, the "on-line learning of hyper parameter" (abstract, L34 and S3.3) is much more appropriately termed as re-estimation of hyper parameter based on mini-batches. This approach to hyper-parameter adjustments is trivial and not interesting --- I suggest that the authors move this part to the appendix and devote more space to improve the clarity of the paper (see points below). The results in sections 4.2 to 4.4 are convincing applications of the proposed approximation. [Clarity] The paper is very unclear in certain critical aspects. 1. L25 --- the term "kernel components" is vague and not defined. 2. The term "boundaries" is used at several places but not well-defined at all. 3. For ADF, the authors should also cite the work by Casto and Opper's Sparse Online Gaussian processes, 2002. 4. L69. The authors need to expand on "scale poorly in long time-series models by still needing to fill the extending domain", since this seems to be an important motivation of the approach. 5. L72. Need to be more precise for "an optimal representation (without approximations) is rewriting". 6. L129. What are the function names in some of the matrix libraries? 7. L155-157: The "convolutional interpolation" needs to be expanded on --- the way it is used here is for $\gamma$, which is different from the way it is applied in [33]. 8. In section 4.1, the details of the experiments, for example, the example functions are not given in the main paper nor the supplementary material. The short forms in Table 1 needs to be explained. Also, are the MAE on the hyperparameters and not the prediction? 9. L225: The text needs to explain more how "Fig. 5 shows that .. reproduce". That is, what is being compared in order to show this? 10. L244: How are the separate learning rates used? [Originality] Section 3 (other than 3.3) is original: the use of the stationary distribution, and the use of DARE to find the stationary solution. [Significance] I deem is work to be of significance, especially when the such streaming data is becoming more common. Minor points [I would have given an 8, if not for so many of these minor points] a. There should be a better wording than saying "data are very long". b. L29 statepaces c. L39: there is be a more appropriate reference for general non-Gaussian likelihoods than [18], which focuses purely on binary classification. d. The word "latent" is used in various places as a noun, when it is an adjective. e. L66. genral f. L68. they they g. Section 2.1 seems to be based almost entirely on [19] or [26] --- suggest the authors to cite these here. h. L118: convereges j. L155: insipired k. L250: This only applies to stationary GPs. [Comments on author's reply] Unfortunately, I have become more confused with the clarification that "kernel components" are the summands in additive kernels, and with the reply the R2 from lines 6 to 11. First, the paper have not given any explicit example or experiments on "additive kernels". Second, I am not sure how much Markovian structure is retained after kernel addition, so the state-space representation may in this case become sub-optimal.

Reviewer 2



This works provides a model for GP inference on univariate temporal series with general likelihoods. Using state-space modeling, the complexity is only O(m^3n) for a state of dimension m and n data points. This is further reduced to O(m^2n) using an infinite-limit approximation. The paper is presented clearly and the authors provide enough information to understand the model in detail. Not many baselines are provided. A relevant reference (and possibly, baseline) is missing: Csató, L. and Opper, M., 2002. Sparse on-line Gaussian processes. Neural computation, 14(3), pp.641-668. The formulation of a time series GP as a state space-model with the corresponding computational savings is not novel. Similarly, using EP (or in this case, ADF) to handle non-Gaussian likelihoods is not new. The novelty of this model is the infinite-limit approximation. This has two effects: a) Decreased computational cost b) Removes the "boundary effect" that increases the uncertainty towards the end of the data There seems to be little practical advantage to a). The computational cost decreases from O(m^3n) to O(m^2n), where m is the dimension of the state space. For practical cases of interest (for instance, the Mattern covariance with nu=3/2), we have m=2. The reduction in computational cost thus seems negligible. This is even more the case when non-Gaussian likelihoods are involved. Table 1 seems to support this understanding. The authors try to make the case for IHGP showing how the gap in computational cost widens as a function of m in Figure 3. However, the need for, say m > 10, is not substantiated, and it might never be required for good practical performance. Effect b), the removal of the "boundary effect", is cast in a positive light at the end of section 3. However, the boundary effect is actually providing a more accurate mean and variance by taking into account the lack of knowledge about the future when making a prediction for the next sample. Therefore, one could expect SS predictions to be more accurate than those of IHGP. This seems to be supported by Table 1 again, where the NLL of the IHGP is consistently higher than that of the SS model. All in all, IHGP seems to sacrifice some accuracy for very little computational gain wrt the SS model, so its practical applicability is questionable, being this the main drawback of an otherwise appealing and well-written work. Minor: Figure placement is not the best throughout the text. It'd be helpful to place figures closer to where they are cited. Many typos throughout the paper, here are some of them: - We are motiviated - While genral-purpose methods - they they scale poorly - In one-dimensonal GPs - the filter gain convereges - stablising - insipired - optimization I have increased my score as a result of the authors' clarifications in their rebuttal.

Reviewer 3



Summary ======= The submission proposes an approximation for Kalman filters that reduces the computational burden w.r.t. the state-space size. Under the condition that observations are equidistant, the state covariance matrix is replaced by the steady-state covariance matrix which allows to simplify the expressions for Kalman gain and filter covariance. With this substitution, certain matrix-matrix multiplications and inversions are no longer required and the complexity reduces from O(nm^3) to O(nm^2) where m is the size of the state space. The authors furthermore show how to apply their approximation when using non-Gaussian likelihoods. To this end they apply EP and demonstrate empirically that a single sweep is enough. Another contribution is a suggestion how to tune hyper-parameters in an online setting. Clarity ======= The writing style is clear. Nothing to complain. My knowledge about Kalman-filters is limited, and so I struggled with Section 3.1 about steady-state filters. I had difficulties to see why the Kalman gain converges until I read elsewhere that this is actually not so easy to show. For readers like me, more references to existing work would be nice (e.g. justification of lines 89 and 118) and references to books (e.g. [5]) should be augmented with page numbers. As far as I understood, Section 3.1. is not novel. This would become clearer if the section would be part of the background. Quality ======= I did not fully understand the paper but the equations appear correct and there is no reason to doubt the results of the experiments. Figure 2 b) shows that the approximation is off initially but quickly becomes almost exact. Is this always the case or it could happen that exact solution and approximation diverge due to the approximation error in the beginning? Originality =========== This submission presents three main ideas: replacing the state covariance matrix by the steady-state covariance matrix, assumed density filtering to cope with non-Gaussian likelihoods and incremental gradient descent for online hyper-parameter tuning. Nothing ground-breaking but definitely novel enough for acceptance. Significance ============ A reduction in complexity without loss of too much precision is a significant achievement.